# Gabapentin Increases Intra-Epidermal and Peptidergic Nerve Fibers Density and Alleviates Allodynia and Thermal Hyperalgesia in a Mouse Model of Acute Taxol-Induced Peripheral Neuropathy

**DOI:** 10.3390/biomedicines10123190

**Published:** 2022-12-08

**Authors:** Michal Klazas, Majdi Saleem Naamneh, Wenhua Zheng, Philip Lazarovici

**Affiliations:** 1Pharmacy Unit, School of Pharmacy Institute for Drug Research, Faculty of Medicine, The Hebrew University of Jerusalem, Jerusalem 9112002, Israel; 2Pharmacology Unit, School of Pharmacy Institute for Drug Research, Faculty of Medicine, The Hebrew University of Jerusalem, Jerusalem 9112002, Israel; 3Center of Reproduction, Development and Aging and Institute of Translation Medicine, Faculty of Health Sciences, University of Macau, Taipa, Macau 999078, China

**Keywords:** Taxol-induced peripheral neuropathy (TIPN), intra-epidermal nerve fibers (IENF), calcitonin gene-related peptides (CGRP), gabapentin (Neurontin), neuropathic pain, analgesia, mice model

## Abstract

The clinical pathology of Taxol-induced peripheral neuropathy (TIPN), characterized by loss of sensory sensitivity and pain, is mirrored in a preclinical pharmacological mice model in which Gabapentin, produced anti-thermal hyperalgesia and anti-allodynia effects. The study aimed to investigate the hypothesis that gabapentin may protect against Taxol-induced neuropathic pain in association with an effect on intra-epidermal nerve fibers density in the TIPN mice model. A TIPN study schedule was induced in mice by daily injection of Taxol during the first week of the experiment. Gabapentin therapy was performed during the 2nd and 3rd weeks. The neuropathic pain was evaluated during the whole experiment by the Von Frey, tail flick, and hot plate tests. Intra-epidermal nerve fibers (IENF) density in skin biopsies was measured at the end of the experiment by immunohistochemistry of ubiquitin carboxyl-terminal hydrolase PGP9.5 pan-neuronal and calcitonin gene-related (CGRP) peptides-I/II- peptidergic markers. Taxol-induced neuropathy was expressed by 80% and 73% reduction in the paw density of IENFs and CGPR, and gabapentin treatment corrected by 83% and 46% this reduction, respectively. Gabapentin-induced increase in the IENF and CGRP nerve fibers density, thus proposing these evaluations as an additional objective end-point tool in TIPN model studies using gabapentin as a reference compound.

## 1. Introduction

Chemotherapy-induced peripheral neuropathy represents a common, dose-limiting, adverse effect of cytotoxic anticancer drug chemotherapy [1]. Taxol (paclitaxel) is one of the most commonly used taxane drugs that specifically inhibit the function of microtubules and, hence, the formation of the mitotic spindle, therefore blocking tumor cell proliferation [2] and angiogenesis [3]. For this reason, Taxol represents a first-line chemotherapy drug in oncology, commonly used to inhibit the progression of ovarian, breast, cervical, endometrial, carcinoma tumors, etc. [4]. Unfortunately, Taxol-induces peripheral neuropathy (TIPN) is an adverse effect, with an incidence of 30 to 50% [5]. It is manifested by loss of cutaneous sensation followed by persistent pain, such as mechanical allodynia (an exaggerated painful response to mechanical stimulation due to axonal degeneration and neurotoxicity to sensory neurons) and thermal hyperalgesia (painful perception of temperature), thus leading to the dose reduction or discontinuation of therapy. Therefore, there is an unmet clinical need to find safe and efficacious drugs for the therapy of TIPN.

To identify possible methods of prevention and treatment for TIPN, it is important to use a validated, preclinical animal model. While several models have been developed, only a few have proven translatable and effective [6]. C57BL/6J mice are commonly used for TIPN research, proving to be most efficacious, quantitatively defined as being reproducible and producing significant peripheral polyneuropathy in at least 90% of the experiments and at least two neurobehavioral outcomes such as thermal hyper- and hypo-algesia and mechanical allodynia [6]. Moreover, C57BL/6J mice are widely used and commercially available, and, in addition to their inbred nature, allow for transgenic and knockout models. The removal of genetic variability minimizes phenotypic or trait variability, thereby enhancing the reproducibility of the model [7]. In the majority of TIPN experiments, the age of the mice ranged from four weeks up to adulthood [8] and TIPN was induced by cumulative doses of Taxol ranging from 4 mg/kg up to 180 mg/kg, delivered in the majority of preclinical studies by the intraperitoneal (i.p.) route of administration, causing a significant peripheral polyneuropathy in 91% of the experiments using males [6]. Mice receiving a Taxol regimen mirror common clinical syndrome of hyperalgesia, allodynia, and numbness as early as the first or third day after receiving their first dose [9]. Low treatment doses guarantee mice welfare, which allows the consistent reproducibility of some clinical manifestations of peripheral neuropathy and their measurement through behavioral assays. Higher doses of paclitaxel result in severe motor deficits and weight loss, obscuring sensory dysfunction and quantification. Therefore, Taxol intermittent low-dose regimen represents the most translatable preclinical dosing protocol in the TIPN mice model. However, it is important to realize that no animal model will represent the full clinical situation perfectly. Nevertheless, the results with mice TIPN models treated with Taxol show high efficacy in causing TIPN (also across sex and various strains used) [6], and this is in line with the findings that Taxol is a chemotherapeutic drug causing one of the highest TIPN rates in patients [10]. With this background, in the present study, we generated the TIPN model in young C57BL/6J male mice by injection of Taxol (6 mg/kg), administered i.p., once daily, for a total of eight injections, to a cumulative intermittent low-dose of 48 mg/kg, to induce prolonged mechanical hypersensitivity and thermal hypoalgesia, mimicking one cycle of chemotherapy treatment in cancer patients [8]. Theoretically, it is preferable to use animal models that replicate all symptoms observed in humans. This remains, however and until today, very challenging. Measures such as numbness, tingling, and ongoing pain rely on verbal reports from the patient, often occur spontaneously, and, therefore, are very difficult to replicate in mice models. Fortunately, the investigations into novel measures of ongoing pain in rodents is an emerging area for research, but, for now, developing animal models of TIPN, which replicate all the symptoms that patients report, remains very challenging, and we, therefore, focused in this study on allodynia/hyperalgesia and intra-epidermal nerve fiber (IENF) density in skin biopsies. Another important issue regarding the clinical relevance of the TIPN model is that the present mice model is tumor-free, whereas, in the clinical situation, most TIPN patients have or experienced previous cancer, which may confound the results related to the use of animal models. We assume that, by using an effective and robust TIPN mice model that mimics the neurobehavioral and pathological clinical situation as much as possible, the translation to the clinical situation will improve in identifying future promising therapies for TIPN.

A recent update of the American Society of Clinical Oncology guideline on the recommended prevention and treatment approaches in the management of chemotherapy-induced peripheral neuropathy in adult cancer survivors reconfirmed that no agents are recommended and suggested that gabapentinoids might be helpful and worth trying for chemotherapy-induced neuropathy [11]. The U.S. Food and Drug Administration (FDA) approve gabapentin (brand name Neurontin) for adjunctive therapy in the treatment of partial seizures and postherpetic neuralgia and various “off-label” (unapproved) uses, including treatment of neuropathic pain caused by diabetic neuropathy, central pain, and TIPN [12], but without solid preclinical research in animal models and clinical trial support [13,14]. In preclinical rodent studies of neuropathic pain, gabapentin was generally used as a positive control to assess dose–response analgesic effects, with a duration of several weeks after frequent administration, and with an effective dose (50%) (ED50) of 27.8 mg/kg [15] with repeated dosing [16]. For example, the analgesic effects of Gabapentin after the induction of chemotherapy-induced neuropathy were observed at a low dose in cisplatin-evoked pain-like behavior, with reduction of mechanical allodynia in mice of both genders that received six i.p. injections of cisplatin (2.3 mg/kg/day) every other day, over the course of two weeks [17]. It has been reported that daily administration of Gabapentin 30 or 100 mg/kg, p.o.(per os, orally), for 14 days [18], 300 mg/kg i.p., twice weekly for 18 days [19], and 100 mg/kg, i.p., representing four daily injections after the peak symptoms [16] suppressed paclitaxel-induced peripheral neuropathy, mechanical hyperalgesia, and allodynia in rat models, respectively. Similarly, a few studies indicated that gabapentin reversed, in C57BL/6 mice, paclitaxel-induced allodynia, with an ED_50_ of 67.4 mg/kg i.p. 0.5–4 h after paclitaxel injection [20] and prevented TIPN at 100 mg/kg i.p., daily injected for eight days in ICR mice [21] and, at 100 mg/kg p.o, injected daily for 14 days in BALB/c mice [18]. However, the effect of gabapentin on intra-epidermal nerve fibers in these rodents’ pharmacological TIPN model using skin biopsies was not yet investigated, representing a significant shortcoming in terms of the relevance of use of Gabapentin as a gold standard compound and of the clinical translational utility of this rodent model.

Intra-epidermal nerve fibers (IENFs) are free nerve endings arising from unmyelinated and thinly myelinated sensory neurons within the skin dermis and are important for the transmission of peripheral pain, allowing the quantification of a small-fiber neuropathy, which may manifest clinically with pain and dysesthesia sensory symptoms, such as burning, stinging and tingling sensations or numbness [22]. Analysis of intra-epidermal nerve fibers (IENFs) in skin biopsy samples has become a standard clinical tool for diagnosing peripheral neuropathies in human patients [23]. A significant reduction in the density of IENFs in the skin contributes to the neuropathic pain in Taxol-induced peripheral neuropathy in rat [24] and mice models [25]. IENF density quantification directly correlates with neurophysiological and neurobehavioral changes and represents, therefore, useful outcome measurement in experimental neuropathy models [26,27,28]. Hence, treatments that protect against chemotherapeutic-induced reduction in the density of IENFs may reduce the development of neuropathic pain and alleviate dysesthesias. Ongoing clinical trials are assessing pharmacological therapeutic strategies to manage chemotherapy-induced peripheral neuropathy, based on preclinical studies in different rodent models [29,30]. However, the relationship between IENF density and TIPN was not yet investigated in mice models under a therapeutic protocol of gabapentin-induced analgesia. Moreover, although the loss of skin innervation in human patients and rodent models of TIPN is now consistently observed, it is yet unknown the extent to which calcitonin gene-related peptides-I/II (CGRP-I/II)- peptidergic nerve fibers were affected during the progressive loss of IENF and if their respective loss is corrected by gabapentin treatment.

Therefore, the present study aimed to assess whether treatment with gabapentin can affect IENFs and CGRP density in the mice’ hind paws at a time point that significantly confers an analgesic effect in an experimental TIPN mice model that closely mimics the course of peripheral neuropathy in human patients. To the best of our knowledge, this is the first study showing the decreased density of CGRP and PGP 9.5-positive nerve fibers innervating the paw skin in the TIPN model and their increase after 21 days of gabapentin treatment, justifying gabapentin use as a gold standard reference compound in the preclinical TIPN mouse model [31,32,33], as found in a randomized, placebo-controlled, clinical trial [34]. Present findings refine the TIPN mice model and further support the claim that skin biopsy with quantification of the density of IENF and CGRP, using generally agreed-upon counting rules, is a reliable, efficient, and objective technique, complementary to analgesia, to assess gabapentin effects on peripheral neuropathy both in patients and in animal models of peripheral neuropathy.

## 2. Materials and Methods

### 2.1. Animals

Young male mice C57BL/6JOlaHsd (four-weeks-old, 18.52 ± 0.32 g at study initiation) were purchased from Envigo RMS Ltd. (Rehovot, Israel) and served as subjects in these experiments, after acclimation for five to seven days to laboratory conditions. During acclimation and throughout the entire study duration, animals were housed within a limited access in a specific pathogen free rodent facility and kept in groups with a maximum of 8–10 mice per cage made of polypropylene (Euro standard type IV, floor area of the cage: 425 × 266 × 185 mm (800 cm^2^)), fitted with solid bottoms and a static cage filter top (Tecniplast Co., Buguggiate, Italy), filled with 7090 Teklad sani-chips animal bedding (Envigo RMS Ltd., Rehovot, Israel), and having two plastic tubes in each cage as enrichment material. During housing, animals were monitored twice daily for health status. No adverse events were observed. During the acclimation period, the mice were assigned to experimental groups in order to reduce possible litter effects. This was conducted using an online random number generator (https://www.graphpad.com/quickcalcs/randomize1/ (accessed on 10 June 2021)) assigning 10 mice subjects to a group. The experiment, including three groups (a. Healthy-Saline, Control; b. Taxol treated-Diseases and c. Taxol-induced disease treated with Gabapentin), was repeated twice (two blocks) during a 12-month period of time. Cages were likewise allocated to treatment groups with randomly generated numbers. Each experimental group was kept in a separate cage to avoid cross-contamination, which can occur through the consumption of fecal material. Mice were provided *ad libitum* with a commercial Teklad 2018S global 18% protein rodent diet (Envigo RMS Ltd., Rehovot, Israel) and had free access to drinking acidified water (using HCl to a pH of 3 ± 0.2) that was monitored periodically and supplied to each cage via polyethylene bottles with stainless steel sipper tubes. Mice were housed in automatically controlled environmental conditions, and the temperature was maintained at 17–23 °C, with a relative humidity of 30–70%, on a 12-h light/dark cycle (light cycle: 7 a.m.–7 p.m; dark cycle: 7 p.m.–7 a.m.) and 15–30 air changes/h in the study room. Random neurobehavioral measurements of 30 mice were blindly conducted between 10 a.m. and 4 p.m. by two female experimenters. If an injection was administered on the same day as the behavior tests, it was administered after all testing had been completed (4–6 p.m.), with the exception of gabapentin and vehicle (saline), which were applied 2 h prior to the neurobehavioral testing. At the end of the study, 54 animals were euthanized by intraperitoneal (i.p.) injection using an overdose of 1000 mg/kg of sodium pentobarbital. The study was twice conducted using 30 mice in each experiment [n = 10 control (healthy, saline), versus n = 10 Taxol (disease), and n = 10 Taxol and gabapentin (disease-gabapentin)]. Two mice before, and four mice during the TIPN study, were excluded due to paw skin injuries that may affect paw biopsies, and, therefore, the neurobehavioral data of these mice were omitted from the data analysis resulting with n = 18 for each experimental group. Two female experimenters were blinded to the allocation of the mice and conduct of the experiments, and another female investigator was blinded for the outcome assessments and data analysis. All procedures were carried out and strictly adhered to the guidelines of the Committee for Research and Ethical Issues of the International Association for the Study of Pain and were approved by an application form (MD-1-1-1027-3117) submitted to the Hebrew University Committee for Ethical Conduct in the Care and Use of Laboratory Animals that approved on 1 September 2020 that the study complies with the rules and regulations set forth. Animal studies (Appendix A) are reported in compliance with the ARRIVE guidelines [35].

### 2.2. Chemicals, Drugs, and Administration

Taxol (paclitaxel) was purchased from both Sigma-Aldrich-Merck (33069-62-4, Saint-Louis, MO, USA), and Tocris (33069-62-4, Minneapolis, MN, USA), and gabapentin was acquired from Thermo Fischer Co. (60142-96-3, Waltham, MA, USA) and Cayman (60142-96-3, Ann Arbor, MI, USA). Sodium chloride physiological solution (saline, 0.9% NaCl) and pentobarbital sodium were obtained from Sigma-Aldrich (57-33-0, Saint Louis, MO, USA). Taxol was dissolved in a solution composed of 50% Cremophor EL and 50% absolute ethanol to a concentration of 25 mg/mL and stored protected from the light at −20 °C, for about 10 days, and then diluted in normal saline (NaCl 0.9%) to a final concentration of 1.2 mg/mL just before administration. Animals in the control healthy group received an injection of vehicle (Cremophor EL: Ethanol, 1:1) diluted in sterile 0.9% NaCl solution to a final concentration of 33.3%. Gabapentin was dissolved in water to a stock solution of 100 mg/mL and, before use, further diluted in saline to a concentration of 30 mg/mL. Animals in the control healthy group received a vehicle saline injection. Drugs and vehicles after filtration on MF-Millipore™ membrane filter (0.22 µm pore size) were administered via the intraperitoneal route (i.p.) in an injection volume of 0.1 mL (about 5 mL/kg body weight). The affinity-purified anti-ubiquitin carboxyl-terminal hydrolase PGP9.5, C-terminal polyclonal antibody produced in rabbits was purchased from Sigma-Aldrich-Merck (SAB1306173, Saint Louis, MO, USA), and recombinant anti-calcitonin gene-related peptides-I/II (CGRP-I/II) antibody [EPR23804-95] (ab283568) and DAPI staining solution (ab228549) were purchased from Abcam, Cambridge, UK). Alexa Fluor 594 affinity pure donkey (red, 112-585-167) and Alexa Fluor 488 goat (green 111-545-144) anti-rabbit IgG-conjugated secondary polyclonal antibody were acquired from Jackson ImmunoResearch Laboratories (West Grove, PA, USA).

### 2.3. Taxol-Induced Peripheral Neuropathy (TIPN)

The present acute TIPN study schedule (Figure 1) was induced in mice by injection of Taxol (6 mg/kg) administered i.p., once daily, from day 0 to 7 (1st week, disease induction), for a total of eight injections, to a cumulative dose of 48 mg/kg. This protocol has been well characterized to produce allodynia [28,36]. Gabapentin and vehicle (saline) were injected 2 h before the Von Frey test on study days 8, 10, 12, 14, 16, and 20 (2nd and 3rd week, therapeutic effect). The neuropathic pain was evaluated by measuring mechanical allodynia using the von Frey test, which was performed in the healthy mice (two days before “−2” for mice adaptation, and one day before “−1” the first Taxol injection, for baseline response) and on the study on days 4, 8, 10, 12, 14, 16, 20, and 21. Mechanical allodynia appeared on the first measurement after Taxol injection (day 4) and persisted for the whole duration of the experiment. The tail flick-test was performed on study day “−1” (baseline, healthy mice) and on days 10, 14, and 21. The hot plate test was performed on study day “−1” (baseline, healthy mice) and days 8, 13, and 20. The body weight of animals was measured three times a week throughout the three weeks study (Figure 1).

### 2.4. Evaluation of Mechanical Allodynia (Von Frey Touch Test)

Allodynia response to tactile stimulation was assessed using the von Frey apparatus (World Precision Instruments, Sarasota, FL, USA) [37] and Electronic Von Frey–e-VF Handheld (product code: 38450, Ugo Basile Co., Gemonio, Italy) [38]. For acclimation to the test environment, the mice were positioned on a metal mesh surface and allowed to move freely. The animals’ cabins were covered with red cellophane to diminish environmental disturbances. The test started after 30 min (cessation of exploratory behavior). The set of von Frey monofilaments provided an approximate logarithmic scale of actual force and a linear scale of perceived intensity. Briefly, von Frey filaments, with approximately equal logarithmic incremental bending forces, were chosen (von Frey numbers equivalent to 0.008, 0.02, 0.04, 0.07, 0.16, 0.40, 0.60, 1.00, 1.40, 2.00, 4.00, 6.00, 8.00, 10.0, 15.0, 26.0 and 60.0 g). When the tip of a fiber of a given length and diameter was pressed against the skin at a right angle and was randomly applied to the left and right plantar surface of the hind paw for 3 s, the force of application increased until the fiber bent. Thereafter, the probe continued to advance, causing the fiber to bend more, but without additional force being applied. The mice exhibited a paw withdrawal reflex of withdrawal, lifting, licking, or shaking the paw, considered a positive response. The minimal force needed to elevate the withdrawal reflex was considered as the value of reference. Decreases in force needed to induce withdrawal were indicative of allodynia, as the force applied is a non-painful stimulus under normal conditions. Evaluations were based on the means ± SEM of mechanical allodynia data. Each treatment group was compared to the vehicle group using statistical analysis.

### 2.5. Evaluation of Thermal Allodynia

#### 2.5.1. Tail-Flick Test

Tail-flick test is a nociceptive essay based on the measurement of the latency of the avoidance response to thermal stimulus in mice [37]. The thermal stimulus was applied on the tail using the Tail Flick Unit–Thermal stimulation (D’Amour and Smith method, product code: 37360, Ugo Basile Co., Gemonio, Italy). When the animal felt discomfort, it reacted with a sudden tail movement. The tail flick or twitch reaction time was then measured and used as an index of animal pain sensitivity. The test was repeated several times on the same animal (with about 5 min resting time between each evaluation), before and after gabapentin administration. In this test, the animals were quiet and immobile during the measurement, which was performed without any holder or restrainer. Evaluations were based on the means ± SEM of thermal allodynia data. Each treatment group was compared to the vehicle group using statistical analysis.

#### 2.5.2. Hot Plate Test

The mice were individually placed on a hot plate (product code: 35300, Ugo Basile Co., Gemonio, Italy), with the temperature adjusted to 55 ± 1 °C. The latency time to withdrawal, shaking, licking the paws, flinching, or jumping to avoid the heat was recorded, and the animal was immediately removed from the hot plate. The time that the mice were left on the plate was limited to 25 s to avoid tissue damage (in case of reduction in the sensory threshold of the animal’s experience) [39]. Evaluations were based on the means ± SEM of thermal allodynia data. Each treatment group was compared to the vehicle group using statistical analysis.

### 2.6. Immunohistochemically (IHC) Staining and Intra-Epidermal Nerve Fiber Counting

Upon experiment termination, 3 mm-diameter punch biopsies were harvested from the hind paw. The tissue was fixed in parafomaldehyde 4%, at 4 °C for 24 hrs., and washed three times with 0.1 M of phosphate-buffered saline (PBS), and stored at 4 °C. Thereafter, it was transferred to sucrose 15% and sucrose 30% in PBS at 4 °C for 24 h until freeze embedding. The samples were freeze-embedded in a perpendicular plane in OCT (optimal cutting temperature compound) medium without decalcifying agents. The frozen tissues were sectioned with 20 μm thickness and stained. After three washings for 5 min in 0.1% Triton-PBS, immunofluorescence staining was performed with primary antibodies at two non-consecutive, free-floating sections using the pan-axonal marker anti-PGP 9.5 and two non-consecutive sections with the peptidergic nerve fiber marker anti-CGRP-I/II. After three washings for 15 min in 0.1% Triton PBS, sections were stained with Alexa Fluor 594 or Alexa Fluor 488 conjugated secondary antibodies, respectively. Finally, after three additional washings for 15 min in 0.1% Triton PBS, the sections were mounted with Fluoromount-G containing DAPI and stored at 4 °C, in the dark, until evaluation. Care was taken to orientate skin sections such that the surface of the epidermis was parallel to the upper limit of the photographed field. The intra-epidermal nerve fiber (IENF) density in the skin was counted using a fluorescent microscope (Olympus BX43, software-cellSens Standard v1, Olympus, Tokyo, Japan), according to published guidelines [26]. Briefly, nerves that branch after crossing the basement membrane were counted as a single unit, and nerves that split or branched below the basement membrane were counted as two units. Nerves that crossed the basement membrane were counted. However, nerve fibers that approached the basement membrane and nerve fragments in the epidermis that did not cross the basement membrane were not counted. IENF number was counted blindly in two sections per 9 samples (n = 9). The IENF density was calculated using the software ImageJ (ImageJ for Mac OS X, version 1.51) as the total number of fibers per unit length of the epidermis (IENF number/tissue in mm), excluding folds, tears, or hair follicles. In addition, peptidergic CGRP positive neuronal fibers were counted in two sections per 4 samples (n = 4).

### 2.7. Statistics

The software GraphPad Prism version 8.0.2 (GraphPad Software Inc., San Diego, CA, USA) was used for plotting graphs, data, and statistical analyses. Statistical analyses were first performed using Shapiro-Wilk for normality test. For multiple comparisons with parametric datasets, the one-way analysis of variance (ANOVA) was performed, and for non-parametric datasets, the Kruskal-Wallis test was performed to test for independence. Dunn’s multiple comparisons post-hoc test was used to analyze differences between specific groups. The results in the figures are expressed as scatter plots with mean ± SEM. The differences were considered significant at * *p* ≤ 0.05, ** *p* ≤ 0.01, *** *p* ≤ 0.001, and **** *p* ≤ 0.0001. Based on prior experience with TIPN assay, and according to a power analysis (JMP, t-test comparison of the mean), in which we have equal size control, and samples with the variables of 26% expected difference in the thermal and mechanical allodynia, 20% expected standard deviation of the data, a desired power of 0.8, and alpha of 0.05, the program predicts a need for ten mice per treatment group.

## 3. Results

### 3.1. Gabapentin Inhibited Mechanical Allodynia and Thermal Hyperalgesia in Taxol-Induced Peripheral Neuropathic Pain

A TIPN study schedule was induced in mice by injection of Taxol administered i.p., once daily, from day 0 to 7 (1st week, disease induction), for a total of eight injections (Figure 1), to a cumulative dose of 48 mg/kg. This treatment was well tolerated, with the exception that mice receiving Taxol (Figure 2A-disease group) were slower to gain weight from days 18 and 21 compared to healthy control mice. However, in the Gabapentin disease-treated group, the mice did not lose the weight (Figure 2A).

Taxol-induced persistent mechanical allodynia was measured up to three weeks after the 1st Taxol injection (Figure 2B). Gabapentin was administered i.p., at a dose of 150 mg/kg, beginning 24 h after the last Taxol injection (Figure 1, 2nd and 3rd week, therapeutic effect), at a cumulative dose of 900 mg/kg, that is, in the range of the doses used to treat chemotherapy-induced, peripheral neuropathy in mice [16,18], reported efficacious for the management of pain in patients [40,41,42], and lower than the maximally tolerated dose in mice of 2000 mg/kg/day [43]. At baseline (day-1, Figure 2B), there were no significant differences between the healthy, disease-vehicle-treated, and disease-gabapentin-treated mice groups. The von Frey mechanical hyperalgesia tests revealed that, compared with the healthy group, the disease-vehicle-treated group showed a very significant decrease in the force needed to induce paw withdrawal reflex, from the 4th to 10th day (from 1.13 ± 0.02 g on the 4th day to 0.35 ± 0.03 g on the 10th day, *p* ≤ 0.0001 vs. healthy, n = 18 in each group), that persisted until 21st day, indicative of peripheral nerve degeneration, neuropathic effect. By contrast, the paw withdrawal reflex threshold was gradually increased in the mice treated with gabapentin from the 10th to 21st day (from 1.23 ± 0.02 g on the 10th day to 1.48 ± 0.03 g on the 21st day, *p* ≤ 0.01–0.001 vs. disease-vehicle, n = 17–18 in each group) (Appendix A) which was interpreted as an analgesic effect.

Using the tail-flick test, at baseline (day-1, Figure 2C), the tail withdrawal latency in the healthy, disease-vehicle treated, and diseased-gabapentin-treated mice groups were similar, with values of 13.05 ± 0.26 sec, 12.50 ± 0.28 sec and 12.83 ± 0.23 sec (n = 18), respectively, thus showing good baseline comparability. Taxol treatment decreased the tail withdrawal latency by about 70% from day 10 to 21 (disease-vehicle group, 3.83 ± 0.22 sec on day 10, *p* ≤ 0.0001, Figure 2C). After administration of gabapentin in the disease-treated group, withdrawal latencies were significantly increased (*p* ≤ 0.01–0.0001) to values of 8.72 ± 0.24 sec, 11.11 ± 0.25 sec, and 12.83 ± 0.31 sec (n = 18), measured on days 10, 14 and 21, respectively (Appendix A). This anti-hyperalgesia effect of gabapentin was also investigated using the hot plate test (Figure 2D). In the hot plate test, the reaction latency of the Taxol treatment group (disease-vehicle), compared to the healthy group, was significantly reduced by 31–48 % (n = 18) on day eight (6.2 ± 0.08 sec; *p* ≤ 0.0001), 13 (5.7 ± 0.09 sec; *p* ≤ 0.0001) and 20 (4.7 ± 0.1 sec; *p* ≤ 0.0001). After administration of gabapentin in the disease-treated group (diseased-gabapentin), withdrawal latencies were significantly increased to values of 7.5 ± 0.15 sec (*p* ≤ 0.01), 8.1 ± 0.09 sec. (*p* ≤ 0.01), and 9.2 ± 0.08 sec (*p* ≤ 0.0001) (n = 18), measured on days 8, 13, and 20, respectively (Figure 2D) (Appendix A). These findings indicate that gabapentin significantly inhibited the Taxol-induced thermal allodynia in the acute thermal pain models of the hot plate and tail-flick tests.

### 3.2. Gabapentin Inhibited Taxol-Induced IENF Loss

IENF loss had been reported in human patients and rodent models of chemotherapy-induced neuropathy [23]. We analyzed the effect of gabapentin treatment on the IENF density by visualization with PGP9.5 immunostaining in the hind paw of Taxol-treated mice three weeks after the first Taxol treatment. Control, healthy mice showed an abundant distribution of nerve fibers entering the epidermis. However, as expected, the IENF density was significantly decreased by about 80% in response to Taxol treatment (disease-vehicle group, Figure 3 and Figure 4). Gabapentin significantly conferred recovery from Taxol-induced IENF loss by increasing intra-epidermal nerve fiber density to about 83% of the healthy mice (Figure 3 and Figure 4) (Appendix A).

Traditionally, the quantification of IENF has relied on immunostaining nerves with the pan-axonal marker, protein gene product 9.5 (PGP 9.5), to measure the density of nerve fibers crossing the basement membrane into the epidermis as a marker of nerve degeneration. No previous studies have quantified peptidergic nerve fiber marker anti-CGRP-I/II in the TIPN mice model to allow the distinction between peptidergic and total IENF concerning neuropathy and gabapentin analgesic effects. Figure 4 indicates that about 25% of IENFs express CGRP-I/II peptidergic marker. CGRP-I/II peptidergic nerve fiber density was significantly decreased by about 73% in response to Taxol treatment (disease-vehicle group, Figure 3 and Figure 4). Gabapentin significantly conferred recovery from Taxol-induced IENF loss by increasing CGRP-I/II to about 46% of the level in the healthy mice (Figure 3 and Figure 4) (Appendix A).

## 4. Discussion

TIPN reflects a serious form of neuropathic pain that correlates with a reduction in the density of IENFs in the skin of cancer patients, which adversely influences treatment. Because of the need for effective pharmacotherapies to treat TIPN that are developed in rodent pharmacological models with well characterized gold-standard drugs, the present study examined whether treatment with gabapentin would affect total (IENF) and peptidergic (CGRP-I/II) nerve fiber density at a time point where it alleviates allodynia and thermal hyperalgesia in a mouse model of acute Taxol-induced peripheral neuropathy.

The findings of this study indicate that gabapentin significantly treated Taxol-induced allodynia and thermal hyperalgesia. Similar, Taxol treatment regimens (multiple injections) and doses (10–50 mg/kg), as used in this study, have been reported to produce painful neuropathy in mice that manifested as thermal hyperalgesia and mechanical allodynia [9,20,21,44]. Present findings of TIPN have been found in other acute and chronic models of peripheral neuropathy, such as mouse models of diabetes [45], HIV-associated distal sensory polyneuropathy [46], anti-retroviral drugs-induced peripheral neuropathy [47], and drug-induced neuropathy [48].

Administration of gabapentin [18,20,49] and other anticonvulsant drugs [50,51] to TIPN rodent models alleviated the development of Taxol-induced allodynia and/or thermal hyperalgesia. In line with these studies, in the present experimental paradigm, gabapentin treatment progressively alleviated Taxol-induced mechanical allodynia and thermal hyperalgesia, as evidenced by increased von Frey force and tail flick and hot plate reaction latencies after two days from the first injection, and for 11–12 days of successive injections (Figure 2), when compared with the healthy and disease-vehicle treated group. These findings are also supported by a study indicating that gabapentin induced significant anti-nociceptive effects compared to other anti-epileptics such as lamotrigine and topiramate [52]. Pregabalin (Lyrica), another γ-aminobutyric acid (GABA) analog, such as gabapentin, also showed a significant increase in tail withdrawal latency and significantly inhibited thermal hyperalgesia in mice [53] and rat [54] TIPN models.

Taxol-induced peripheral neuropathy is associated with degeneration or loss of intra-epidermal nerve fibers (IENFs) that have been associated with thermal hyperalgesia and mechanical allodynia [55,56,57]. It is plausible that this reduction in the density of IENFs in the skin contributes to neuropathic pain in TIPN mice models [25]. IENF density quantification directly correlates with neurophysiological and neurobehavioral changes and represents, therefore, useful outcome measurement in experimental neuropathy models [26,27,28]. The principal finding in the current study is that, in an acute TIPN mice model, there is a significantly decreased proportion of IENFs, including those immunolabelled for CGRP-I/II. To the best of our knowledge, this is the first study to identify quantitative changes in the CGRP-I/II nerve fibers, in addition to IENF, that might be important in the development of Taxol-induced neuropathy and Gabapentin-induced analgesic effect. The decrease in the number of CGRP-immunolabelled nerve fibers could be the result of enhanced CGRP release or a loss of nerve fibers. The decrease in the PGP 9.5-immunoreactive and the double-labeled CGRP and PGP 9.5 nerve fibers in the same anatomical area may indicate a degenerative loss of nerve fibers in the skin and their regeneration after Gabapentin treatment. The IENFs are considered to relay pain information to the central nervous system and have been divided into two distinct groups based on neurochemical criteria: peptidergic fibers containing one or more neuropeptides, such as CGRP, substance P, somatostatin, and the remaining nerve fibers are non-peptidergic. The question of whether the density of the non-peptidergic nerve fibers is also affected by gabapentin deserves further investigation. Present results suggest that CGRP-I/II-peptidergic nerve fibers as total IENF are significantly lost in the mouse epidermis after TIPN, and increased after 21 days of Gabapentin treatment, in parallel to behavioral deficits, suggesting that the neurobehavioral symptoms and their alleviation may be tied to damage within the peptidergic and non-peptidergic nerve fibers. Quantitation of IENF is proposed as a more objective tool for the evaluation of peripheral neuropathy in animal models of TIPN and other peripheral neuropathies.

The precise molecular mechanism of mechanical allodynia and sensitivity to heat observed upon Taxol treatment are not clearly understood. The present working hypothesis claims that Taxol probably causes dysfunctional microtubules in dorsal root ganglia, axons, and Schwann cells, resulting in dysfunction in calcium signaling, neuropeptide, growth factor release, mitochondrial damage, reactive oxygen species formation, and ion channels, cumulatively contributing to the neuropathic pain [36]. Previous experimental studies had shown that Gabapentin binds with a high affinity to the α2δ1-subunit of the different types of voltage-dependent calcium channels [58], known to interact with N-methyl-d-aspartate-sensitive glutamate receptors, neurexin-1α, thrombospondins (adhesion molecules), and other presynaptic proteins [59], thereby reducing the release of both excitatory and inhibitory neurotransmitters from the rat spinal cord dorsal horn [60]. This possible mechanism could account in part for the anti-nociceptive effect of gabapentin. The ability of gabapentin to restore IENF and CGRP I/II-peptidergic nerve fibers density in the TIPN model maybe be explained by an increased level of skin neurotrophins, such as nerve growth factor (NGF), known to induce sprouting of cutaneous sensory nerve fibers [61]. Indeed, several studies have implicated NGF in gabapentinoids’ analgesic and IENF effects in rat diabetic skin [61,62], a possibility deserving further investigation. Moreover, gabapentinoid treatment promoted corticospinal axon regrowth ability following murine spinal cord injury [63], indicating an ability to induce nerve fiber sprouting.

In conclusion, the results from this study confirm that, in the acute TIPN mice model, Taxol-induced peripheral neuropathy is expressed by allodynia and thermal hyperalgesia. Gabapentin alleviated these effects, in parallel to the increase in IENF and CGRP I/II-peptidergic nerve fibers density, refining the TIPN model and indicating that it can be used as a gold standard reference in the TIPN mice model for preclinical research and development of drugs. We assume that further quantification of the different types of IENF innervations in mice footpads in studies with the TIPN mice model will become a useful popular tool and will offer objective quantification metrics in addition to the animal neurobehavioral outcomes related to pain. Skin biopsies might be useful in detecting early changes of IENF density, which predict the progression of neuropathy and assess degeneration and regeneration of IENF. However, further TIPN studies using mice of different gender, age, and genetic backgrounds are warranted to confirm the potential usefulness of skin biopsy with the measurement of IENF/ CGRP I/II nerve fiber density as an outcome diagnostic and/or therapeutic measure in preclinical, pharmacological rodent models research.

## Figures and Tables

**Figure 1 biomedicines-10-03190-f001:**
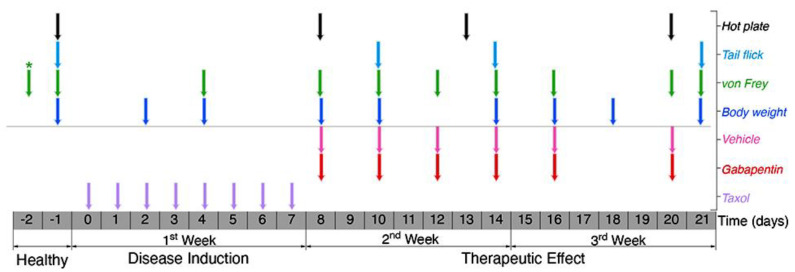
The schematic presentation of the TIPN study protocol. A TIPN was induced in mice (n = 40) by daily injection of Taxol from day 0 to 7 (1st week, disease induction) to a cumulative dose of 48 mg/kg. Gabapentin and vehicle saline were injected during the 2nd and 3rd week (therapeutic effect). The neuropathic pain was evaluated by measuring mechanical allodynia using the von Frey test in the healthy mice (two days (*) and one day before the first Taxol injection), for baseline response and during the disease and therapy period, and thermal hypersensitivity was measured four times each, with the tail-flick and the hot plate tests during the entire protocol. The body weight of animals was measured three times a week throughout the entire protocol. Healthy represents the mice cohort before TIPN. Colored arrows represent the day of drug delivery and performance of the neurobehavioral test.

**Figure 2 biomedicines-10-03190-f002:**
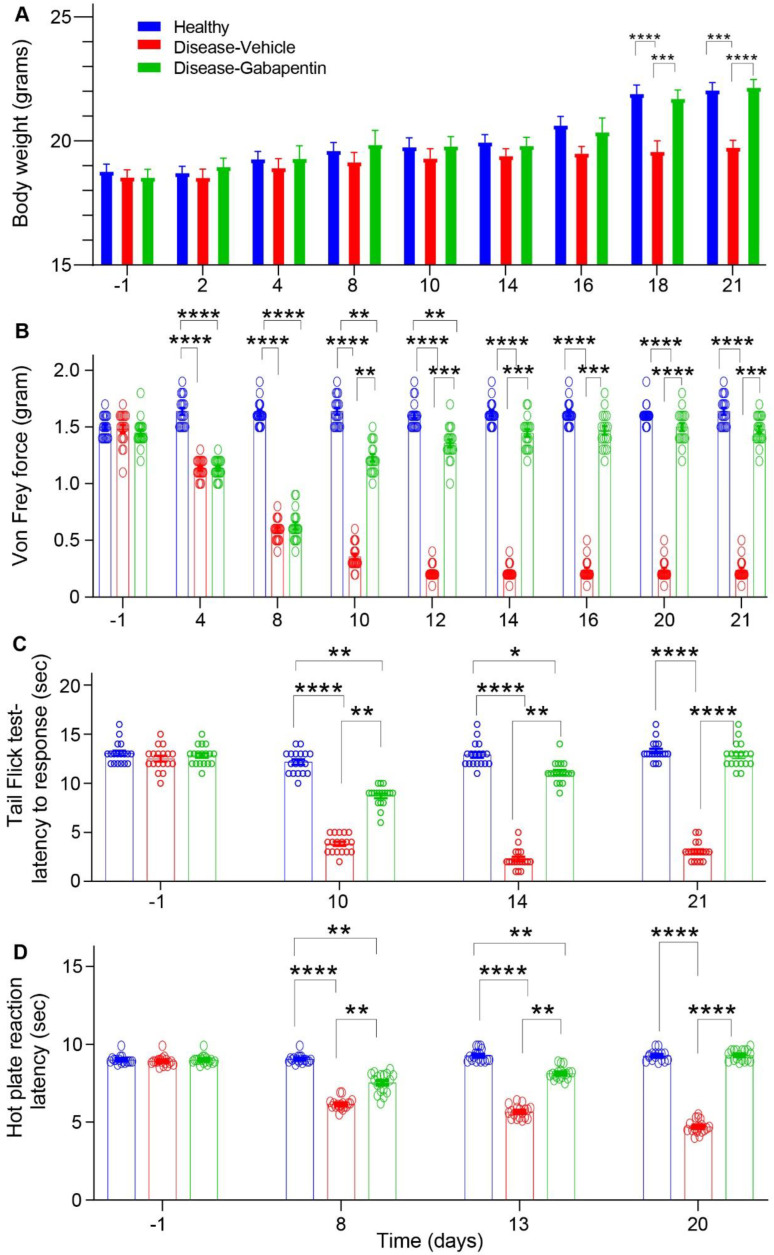
The inhibitory effects of gabapentin on the body weight and on Taxol-induced mechanical and thermal allodynia in mice. (**A**) Taxol-treated mice (disease-vehicle) gained less weight than controls (healthy) from day 18–21, but their weight normalized in gabapentin-treated mice (disease-gabapentin). (**B**) Taxol-induced hypersensitive response to von Frey filament stimulations test (disease-vehicle) compared to control (healthy). Taxol-induced mechanical allodynia was significantly improved during two weeks of gabapentin treatment (disease-gabapentin). (**C**) Tail-flick test latency to response was significantly decreased in Taxol-treated mice (disease-vehicle) compared to control (healthy), but it was normalized in gabapentin-treated mice (disease-gabapentin). (**D**) The reaction latency times measured using the hot plate thermal hyperalgesia test were significantly decreased in Taxol-treated mice (disease-vehicle) compared to control (healthy), but they were normalized in gabapentin-treated mice (disease-gabapentin). The results are means ± SEM (n = 18); The level of significance (*) assumed was 1% (** *p* ≤ 0.01); 0.1% (*** *p* ≤ 0.001); 0.01% (**** *p* ≤ 0.0001); one-way ANOVA, followed by Dunn’s post-test.

**Figure 3 biomedicines-10-03190-f003:**
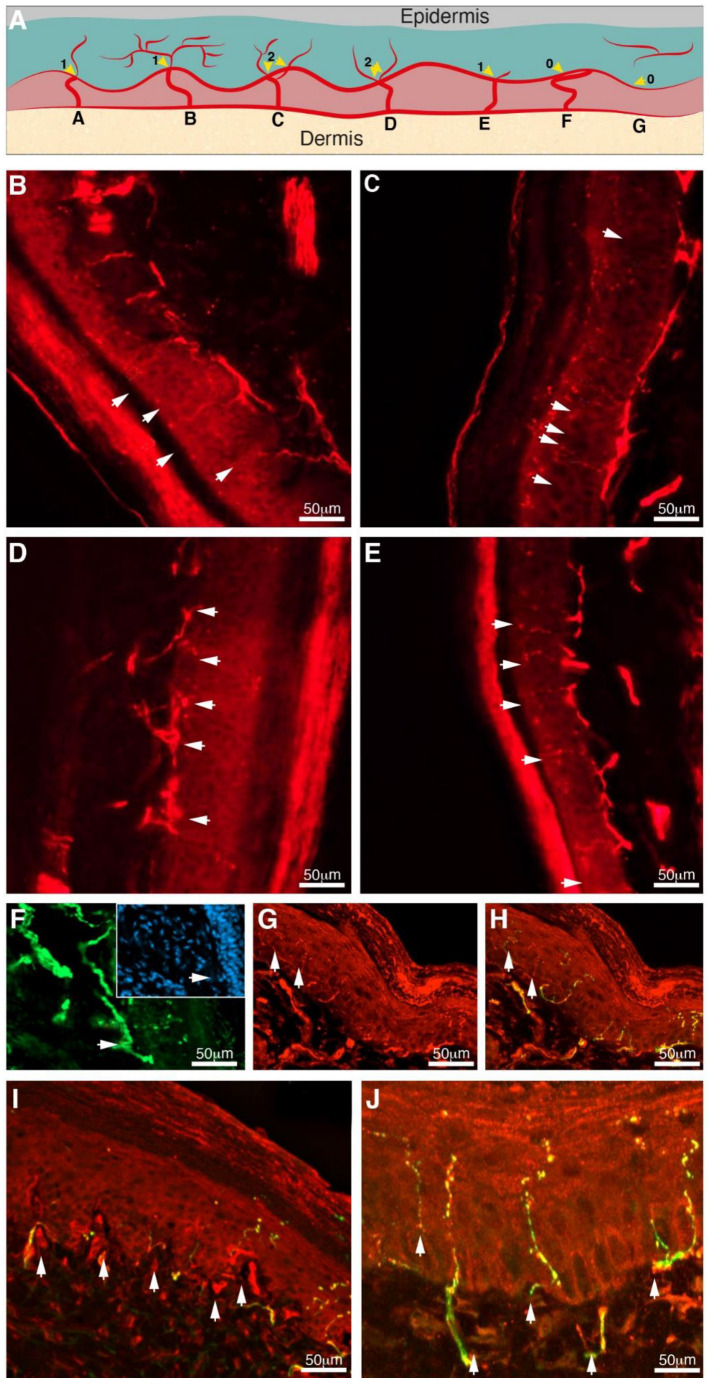
A schematic diagram depicting the intra-epidermal nerve fiber (IENF) counting rules and immunohistochemistry photos of punch biopsies of the different experimental groups. (**A**) Diagram of skin innervation with IENF estimation rules that are widely used in human studies: nerves (red lines), basement membrane (pink area), dermis (yellow), and epidermis (blue-gray shade). (A) Count as a single unit nerve (score 1) that crosses the basement membrane of the epidermis; (B) Nerves that branch after crossing the basement membrane are also counted as a single unit (score 1); (C) Nerves that split below the basement membrane are counted as two units (score 2); (D) Nerves that appear to branch within the basement membrane are counted as two units (score 2); (E) Nerve fragments that do cross the basement membrane are counted as one unit (score 1) (F) Nerve fibers that approach the basement membrane but do not cross it are not counted (score 0); (G) Nerve fragments in the epidermis that do not cross the basement membrane in the section are not counted (score 0); (**B**) Healthy mice—white arrows indicate nerves labeled with anti-PGP 9.5 antibody (red, n = 9); (**C**) Taxol-treated mice (disease-vehicle)—white arrows indicate nerves (n = 9); (**D**,**E**) Gabapentin-treated mice (disease-vehicle)—white arrows indicate nerves (n = 9); (**F**) CGRP-positive nerves (green, n = 4) represent peptidergic IENFs labeled with white arrows; Insert: DAPI labeled nuclei in blue; (**G**) IENFs immunostained for PGP 9.5 (red), representing all epidermal nerve fibers labeled with white arrows; (**H**) Merged F and G view; (**I**). Taxol-treated mice (disease-vehicle; n = 4)—white arrows indicate peptidergic IENFs; (**J**) Gabapentin-treated mice (disease-vehicle; n = 4)—white arrows indicate peptidergic IENFs.

**Figure 4 biomedicines-10-03190-f004:**
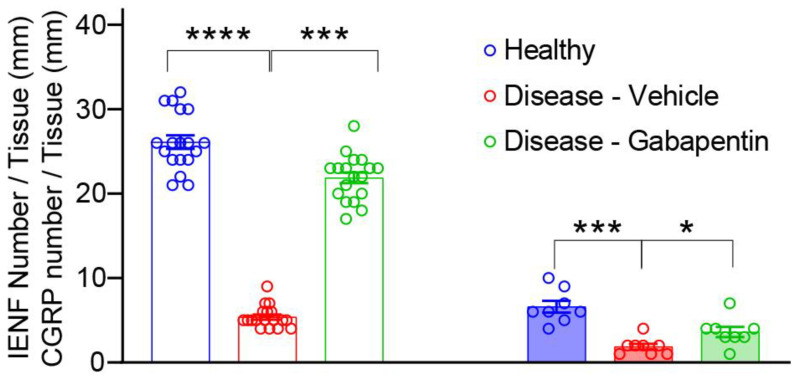
Quantification of immunohistochemically staining of IENF (left) and CGRP-I/II (right) density in skin biopsies of healthy, Taxol-treated (disease-vehicle) and gabapentin-treated mice (disease-gabapentin). The epidermal total (IENF) and CGRP-peptidergic nerve fiber innervation density (mean +/− SEM) on day 21 were measured and expressed as the number of IENF (left bars) and CGRP (right bars) per length of the section (IENF/mm, n = 9; CGRP/mm, n = 4); The level of significance (*) assumed was 5% (* *p* ≤ 0.05); 0.1% (*** *p* ≤ 0.001); 0.01% (**** *p* ≤ 0.0001); one-way ANOVA followed by Dunn’s post-test.

## Data Availability

The data presented in this study are available in the article and Appendix A.

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
