# Peer review of "Gabapentin Increases Intra-Epidermal and Peptidergic Nerve Fibers Density and Alleviates Allodynia and Thermal Hyperalgesia in a Mouse Model of Acute Taxol-Induced Peripheral Neuropathy"

_biomedicines, 2022, doi:10.3390/biomedicines10123190_

Round 1

Reviewer 1 Report (Previous Reviewer 2)

The resubmitted form of paper titled (Gabapentin increases intra-epidermal and peptidergic nerve fibers density and alleviates allodynia and thermal hyperalgesia in a mouse model of acute Taxol-induced peripheral neuropathy) by Klazaz et al. was partly improved

I suggets the authors send also a new clean form (without tracj changes) to be able to follow 

1- Write the applied test in each figure legend

2- write the n in each figure leegnd

Author Response

Reviewer 2 Report (Previous Reviewer 1)

The authors have gracefully made most of the changes I requested and provided solid, good reason not to conduct the evaluation of IENF density for non-peptidergic sensory terminals.

My only additional request is that they consider using thinner border lines in the scatter bar-dot plots, as in its present form the circles representing the actual data points are very difficult to see.

Round 2

Reviewer 1 Report (Previous Reviewer 2)

thanks for the corrections

Author Response

Your understanding is appreciated!

This manuscript is a resubmission of an earlier submission. The following is a list of the peer review reports and author responses from that submission.

Round 1

Reviewer 1 Report

This short and straight-forward study by Rubin et al. presents some behavioural evidence that i.p. administration of repetitive therapeutic doses of gabapentin significantly alleviate mechanical and thermal allodynia in a mice model of taxol induced peripheral neuropathy. The authors examine a single possible causative event for this action of the gabapentinoid, namely changes in the epidermal nerve fiber density in all experimental groups at the end of the study.

Albeit the overall rationale of the study is sound and the data are fairly consistent with the driving hypothesis of the authors, the study is too superficial and it would be greatly improved by the addition of extra data.

First, and foremost, the initial decrease in IENF described in rodents and humans under taxol treatment is linked to hyposensitivity (not hypersensitivity as stated in the discussion lines 329-330). This is due to the reduction in the number of fibers, and thus, to absence of free nerve terminals capable of sensing any cunateously appplied stimulus. It is later on that taxols induce persistent pain, and the underlying mechanisms involve aberrant spontaneous neuroregeneration, alteration in redox status, up-regulation of sodium channels, etc. To assume that just restoring innervation reverses allodynia requires more functional proof.

The authors must improve figure 2 substantially by adding a counterstain with DAPI (or a nuclear marker) thus allowing a clear separation between the different skin layers. They must also stain the nerve terminals in order to ascertain what neuronal population is being affected preferentially by taxol and what is being rescued by the gabapentinoid. In this respect, I would suggest using a marker of peptidergic terminals (i.e. substance-P or CGRP), and two more closely associated to thermal sensitivity (TRPV1 and TRPM8, for example). It would also be very valuable to see if IB4-binding, non-peptidergic neurons are also affected. This is essential to at least have some idea of what type of fibers are involved and make a really novel contribution to this field.

As a minor point, I would like to know how many mice had to euthanised as a result of the taxol treatment.

Have the authors used power calculations to determine the size of each experimental group? What randomisation algorithm, if any, was used?

I also expect the authors to show bar-scatter plots instead of solid bars, as this is more transparent to show the true magnitude of the average inidivual variations in the various behaviours and IENF densities reported here.

Finally, why have the authors limited the study to male mice? Taxol is used to treat especially gynaecological cancers (endomentrial and breast cancer) and thus, including females would have been truly interesting.

Reviewer 2 Report

paper titled (Gabapentin reversed allodynia and thermal hyperalgesia in direct correlation with the increase of intra-epidermal nerve fibers density in a mouse model of acute Taxol-induced peripheral neuropathy) by Rubin et al. This is a good trial but of curse NOT the first one to explore the effect of gabapentin on taxol-induced NP. Authors need to strongly explore the novelty of the study in Absc=tract, introduction and conclusion. & also to follow the following recommendations as the paper needs extensive revision. It would be great if the authors have data about the effect of gabapentin in normal animals.

1- Title: needs revision for the word "reverse" as whether gabapentin completely reversed the situation to the normal state? confirm if you think so

2- Abstract: should be amended by some numerical values

3-Aim of the study is not completely shown at the end of introduction 

4-What was the age of the animals at the breginning of the experiment

5- Give the method of solubilizing or dilution of the drugs.

6- Figure 1 should be divided into 2 figures: one for the experimental design in methods & 1 in results 

7- 2.3. Taxol-induced peripheral neuropathy (TIPN): should be divided into 2 parts : one for the model & parts related to pain tests should be moved to each test

8- write the model and origin of each apparatus

9- How long tissues remained in sucrose? did they need decalcification?

10- Authors have to check the normality of distribution of the results by a suitable post hoc test (such as Shapiro-Wilk test or K-S test) before deciding to choose certain ANOVA. If the normality test indicated normal dist of the data, so use one-way ANOVA, if not, use non parametric ANOVA. In all cases choose a suitable post-hoc test

11- Data should be presented as mean+-SD (not SE) this is as authors do not cover the universe for this study.

12- Authors should give the source of chemicals, kits and antibodies completely and consistently (code, company, town, state and country) & version for software.

13- In each illustration mention the type of the presented data & the statistical test applied for analysis 

14- Methods in general lacks references at many occasions

15- Figure 2 should be moved to results

16-     Use appropriate abbreviations for minutes, seconds...etc

17-     Mention "n" in each illustation individually

18- Give refernces for gabapentin doses